# Optimisation and Validation of a Nutritional Intervention to Enhance Sleep Quality and Quantity

**DOI:** 10.3390/nu12092579

**Published:** 2020-08-25

**Authors:** Shona L. Halson, Gregory Shaw, Nathan Versey, Dean J. Miller, Charli Sargent, Gregory D. Roach, Lara Nyman, James M. Carter, Keith Baar

**Affiliations:** 1School of Behavioural and Health Sciences, Australian Catholic University, Banyo QLD 4014, Australia; 2High Performance Unit, Swimming Australia, Brisbane QLD 4519, Australia; greg.shaw@swimming.org.au; 3Rowing Australia, Yarralumla ACT 2600, Australia; nathan.versey@rowing.ausportnet.com; 4Appleton Institute for Behavioural Science, Central Queensland University, Wayville 5034, Australia; dean.miller@cqu.edu.au (D.J.M.); charli.sargent@cqu.edu.au (C.S.); greg.roach@cqu.edu.au (G.D.R.); 5Gatorade Sport Science Institute, PepsiCo Global Research and Development-Life Sciences, Purchase, NY 10577, USA; lara.nyman@pepsico.com (L.N.); james.carter@pepsico.com (J.M.C.); 6Departments of Neurobiology, Physiology & Behavior and Department of Physiology and Membrane Biology, University of California Davis School of Medicine, Davis, CA 95817, USA; kbaar@ucdavis.edu

**Keywords:** nutrition, polysomnography, sleep onset latency

## Abstract

Background: Disturbed sleep may negatively influence physical health, cognitive performance, metabolism, and general wellbeing. Nutritional interventions represent a potential non-pharmacological means to increase sleep quality and quantity. Objective: (1) Identify an optimal suite of nutritional ingredients and (2) validate the effects of this suite utilising polysomnography, and cognitive and balance tests. Methods: The optimal and least optimal combinations of six ingredients were identified utilising 55 male participants and a Box–Behnken predictive model. To validate the model, 18 healthy, male, normal sleepers underwent three trials in a randomised, counterbalanced design: (1) optimal drink, (2) least optimal drink, or (3) placebo were provided before bed in a double-blinded manner. Polysomnography was utilised to measure sleep architecture. Cognitive performance, postural sway, and subjective sleep quality, were assessed 30 min after waking. Results: The optimal drink resulted in a significantly shorter sleep onset latency (9.9 ± 12.3 min) when compared to both the least optimal drink (26.1 ± 37.4 min) and the placebo drink (19.6 ± 32.0 min). No other measures of sleep, cognitive performance, postural sway, and subjective sleep quality were different between trials. Conclusion: A combination of ingredients, optimised to enhance sleep, significantly reduced sleep onset latency. No detrimental effects on sleep architecture, subjective sleep quality or next day performance were observed.

## 1. Introduction

Sleep has important biological functions in a myriad of physiological processes including learning, memory, and cognition [1,2]. Restricting sleep to less than 6 h per night for four or more consecutive nights has been shown to impair cognitive performance and mood [3], disturb glucose metabolism [4], appetite regulation [5], and immune function [6]. Common prescription medications for insomnia may result in adverse side effects, including tolerance and addiction, rebound insomnia upon cessation, and reduced emotional and cognitive function with long-term use [7]. Furthermore, pharmaceutical sleep interventions may result in next-day psychomotor impairment [8]. Therefore, novel interventions, as alternatives to pharmacological interventions, are needed.

A number of neurotransmitters are associated with the sleep–wake cycle, and include: serotonin (5-hydroxytryptamine; 5-HT), gamma-aminobutyric acid (GABA), orexin, melatonin, galanin, noradrenaline, and histamine [9]. Dietary interventions that act upon these neurotransmitters in the brain and their role in changing sleep quality and quantity have become of interest to those looking to improve sleep without the use of pharmaceutical intervention.

Dietary precursors can influence the rate of synthesis and function of a small number of neurotransmitters, including serotonin [10]. Synthesis of serotonin (5-HT) may influence sleep and is dependent on the availability of its precursor, the amino acid L-tryptophan (Trp), in the brain. Trp is transported across the blood–brain barrier by a system that shares other transporters including several large neutral amino acids (LNAA). Thus, the ratio of Trp/LNAA in the blood is crucial to the transport of Trp into the brain. Since Trp is the least abundant amino acid in food proteins, ingestion of many protein-rich foods generally decreases the uptake of Trp into the brain, due to the relative rise and preferential transport of LNAAs into the brain [10]. However, carbohydrate (CHO) intake increases brain Trp as a net result of the action of insulin. Insulin stimulates the uptake of LNAA into skeletal muscle (increasing the ratio of Trp:LNAA in the blood), but also reduces blood concentrations of free-fatty acids, which results in increased binding of Trp (reducing blood levels of free Trp) [11]. Additionally, ingestion of specific protein fractions, high in Trp, have been show to enhance Trp availability and improve sleep related measures [12,13].

Dietary supplements that have some evidence of positively influencing sleep include: CHO, α-lactalbumin, tart cherry juice, valerian, L-theanine, and nucleotides [14,15,16,17,18]. Intake of these factors represents a potential intervention for enhancing natural sleep. A number of recent reviews have highlighted the potential for a number of these ingredients to influence sleep and jetlag [16,19]. However, each of these reviews make mention of the need for additional randomised controlled trials, using objective measures of sleep to enhance the quality of research in the area. Since there are multiple mechanisms, both known and unknown, by which nutrition may affect sleep, we hypothesised that a combination of nutrients would be more effective than any of the single nutrients studied to date. Furthermore, since these agents have only been tested in isolation, we hypothesised that there would be interactions between different sleep promoting pathways. Therefore, is was necessary to perform an experiment that would allow us to simultaneously determine the effect of all six nutrients at one time, regardless of mechanism and bias.

The aim of this study was to (1) identify the optimal combination and dosage of this suite of nutritional ingredients and (2) validate the optimal ingredient mix identified using gold standard sleep monitoring (polysomnography) by determining the effects of supplementation on sleep quantity and quality as well as subjective sleep, balance, and cognitive function in healthy adults.

## 2. Methods—Optimisation of Nutritional Intervention

Fifty-five healthy males (Age: 27.4 ± 6.2 year; Weight 83.2 ± 11.5 kg; Height; 181.2 ± 6.7 cm) attended the laboratory on two occasions (one screening plus baseline session and one testing session). Participants were provided with a standardised diet controlling for caffeine, total, carbohydrate (40%), and protein (30%) for the 24 h prior to visit 2. To determine the optimal concentration and combination of the identified nutritional ingredients, a six-factor Box–Behnken model was produced using Design-Expert software (Stat-Ease, Inc., Minneapolis, MN, USA). The Box–Behnken design (a subset of design of experiment methodology) is an incomplete factorial design that uses three levels of each factor being tested and analyses the resulting outcome data for effects, via quadratic response surface plots. According to the Box–Behnken design, each subject was provided with one intervention, making for 55 independent trials (48 intervention trials, plus 7 centre point trials). The 7 centre point trials consisted of participants being provided the mid-range dose for each ingredient. The seven participants who received the central combination of nutrients were used to determine the biological variability of the measures, and provide the statistical basis for the optimal selection. As each participant completed one trial only, the subjects were allocated a supplement in the order they were recruited. The study was approved by the Australian Institute of Sport Ethics Committee (Approval number: 20131003) and all participants completed written informed consent (ClinicalTrials.gov Identifier: NCT03288077).

### 2.1. Dietary Supplement

The dietary supplement was given at time point 0 and blood samples taken at 0 (pre-drink ingestion), 30-, 60-, 90-, 120-, 150-, and 180-min post ingestion of the supplement. The drink provided to the participants was a combination of the below ingredients:Tart cherry juice (0 ml, 50 mL, 100 mL) (Cherry Active Australia, Sydney, Australia)High GI CHO (0 g, 25 g, 50 g) (PolyJoule, Nutricia, Sydney, Australia)α-lactalbumin (0 g, 20 g, 40 g) (Davisco Foods, Le Sueur, MN, USA)Adenosine-5-monophosphate (5-AMP)-(0 mcg, 26.5 mcg, 53 mcg) (Sigma, Ronkonkoma, NY, USA)Valerian (0 mg, 750 mg, 1500 mg) (Martin Baeur Group, Vestenbergsgreuth, Germany)Theanine (0 mg, 500 mg, 1000 mg) (Suntheanine, Osaka, Japan)

### 2.2. Assessment of Effectiveness

Free tryptophan was assessed using ultra-performance liquid chromatography tandem mass spectrometry (UPLC-MS/MS) at each time point (baseline and every 30 min until 180 min). A three-choice vigilance task (3CVT) lasting 5 min was completed at each time point (baseline and every 30 min until 180 min). The 3CVT requires participants to discriminate one primary target (presented 70% of the time) from two secondary non-target geometric shapes that are randomly interspersed over the 5-min test period. Each 3CVT consists of a single stimulus type (target or non-target) presented for 0.2 s, with fixed timing between stimuli. Participants were instructed to respond as quickly as possible to each stimulus presentation by selecting the left arrow to indicate target stimuli, and the right arrow to indicate non-target stimuli. Performance measurements include reaction time and accuracy (i.e., percentage of correct responses). Participants were administered a “practice round” before beginning the 3CVT to ensure that the participants fully understood the requirements of the task.

### 2.3. Model Optimisation

The Box–Behnken model was optimised for 3CVT and blood tryptophan levels, since these measures are believed to be quantitative physiological predictors of sleepiness. Following data collection, the 3CVT and tryptophan measures were fed back into the Design-Expert software and a model of treatment versus response was produced. This model was then used to predict the optimal and least optimal combination and concentration of each of the ingredients. The model was validated in a separate cohort of participants using sleep monitoring (see below).

## 3. Methods—Validation

### 3.1. Design and Procedures

Eighteen healthy males participated in the validation component of the study (Age: 25.1 ± 6.2 year; Weight 71.6 ± 11.3 kg; Height; 176 ± 8.3 cm). None of the participants had a history of a sleep disorder and none were taking medication during the time of testing. Participants were all non-smokers. None of the participants had undertaken transmeridian travel or shift work in the month prior to participation in the study. Over the previous month, participants’ self-reported bedtime was 23:12 ± 00:54 h; self-reported get-up time was 08:12 ± 01:18 h; self-reported sleep onset latency was 15.0 ± 10.2 min and self-reported total sleep time was 8.1 ± 1.1 h. Participants were excluded if their usual self-reported bedtime was later than 01:00 h or their usual self-reported get-up time was later than 10:00 h. On average, over the past month, participants consumed 1.0 ± 0.9 caffeinated beverages per day and consumed 1.6 ± 1.1 units of alcohol per day (each unit equivalent to 14 g of pure alcohol). There were no participant dropouts throughout the study.

A double-blind, placebo-controlled crossover experimental design was employed. Participants attended the Appleton Institute Sleep Laboratory and completed overnight polysomnography on four consecutive nights. The first night served as an adaptation night and was used to familiarise participants with the equipment for monitoring sleep. The next three nights were experimental nights on which participants received one of three interventions in a randomised, counterbalanced order: (1) optimal drink, (2) least optimal drink, or (3) placebo. On each night, participants consumed the drink at 21:00 h and were instructed to consume each drink within 5 min. Electrodes for monitoring sleep were then applied to participants. Participants were given a 9.5-h sleep opportunity from 22:30 h to 08:00 h. Participants rated their subjective sleepiness level every thirty minutes from 20:00 h until 22:30 h. Participants were not permitted to consume any water after 21:00 h. At 08:30 h the following morning (i.e., 30 min after waking), participants rated their subjective sleepiness, sleep quality, sleep duration, sleep latency, and completed a gastrointestinal symptom scale. At 09:00 h, participants completed a test battery to assess subjective alertness, self-perceived capacity, cognitive performance, and postural sway as outlined below. The test battery was always conducted in the same order and took approximately 30 min to complete.

### 3.2. Living Conditions

Participants were housed in a purpose-built accommodation suite at Central Queensland University’s Appleton Institute. The suite, configured like a serviced apartment, can accommodate six participants concurrently, each with a private bedroom, lounge room, and bathroom. During scheduled daytime wake episodes, participants were free to engage in activities such as reading, watching movies, and listening to music. Participants were not permitted to sleep outside of scheduled time in bed. Researchers monitored participants for compliance in person and via CCTV. At the end of the protocol, participants were provided with transport to their home to minimise any risk of driving if sleep was at all disturbed.

### 3.3. Meals

All meals provided to participants were calorie-controlled and the participants consumed the same approximate number of calories at each meal. The participants were served breakfast, lunch, and dinner at 09:30 h, 12:30 h, and 19:00 h, respectively. An afternoon snack was also provided at 16:00 h. On average, participants consumed 10,061 ± 1112 kJ per day containing 51.8 ± 6.2% carbohydrate, 21.7 ± 6.5% protein, and 23.8 ± 6.7% fat. Participants did not have access to any other food or beverages (other than water) outside of the designated meal/snack times and were not permitted to consume caffeine or alcohol at any time during the protocol.

### 3.4. Sleep Assessment

Sleep was recorded using polysomnography equipment (Grael; Compumedics, Melbourne, VIC, Australia) with a standard montage of electrodes. Electrodes were applied in the 60 min prior to lights out and included three electroencephalograms (C4-M1, F4-M1, O2-M1), two electrooculograms (left/right outer canthus), and a submental electromyogram. All sleep records were blinded and manually scored in 30-s epochs by the same technician according to established criteria [20]. Stages of sleep were identified as non-rapid eye movement sleep (stages N1, N2, N3) and rapid eye movement sleep (R), with N1 considered the lightest phase of sleep and N3 considered the deepest phase. The following dependent variables were calculated from each sleep recording: total sleep time (min), the time spent in any stage of sleep (i.e., N1, N2, N3, R) during time in bed; time spent in stages N1, N2, N3 and R sleep (min); sleep onset latency (min), the time between lights-out to the first epoch of any stage of sleep (i.e., N1, N2, N3, R); wake after sleep onset (min), the time spent in bed awake minus sleep onset latency; sleep efficiency (%), total sleep time divided by time in bed x 100; arousals, (count); arousals in NREM (count); arousals in REM (count); awakenings (count); stage shifts (count); stage R onset latency (min); and stage N3 onset latency.

### 3.5. Subjective Sleepiness

Subjective sleepiness was assessed using the Karolinska sleepiness scale (KSS) [21]. The KSS is a 9-point scale where 1 = “extremely alert”, and 9 = “very sleepy, great effort to keep awake, fighting sleep”. Participants were instructed to circle the number on the scale that corresponded to their current level of sleepiness.

### 3.6. Subjective Sleep Quality, Sleep Duration, Sleep Latency

Sleep quality was assessed using a 7-point scale, where 1 = “extremely poor”, 2 = “very poor”, 3 = “poor”, 4 = “average”, 5 = “good”, 6 = “very good”, and 7 = “extremely good”. Participants were verbally asked “how much sleep do you think you got?” and “how long did it take you to fall asleep?”.

### 3.7. Gastrointestinal Symptom Scale

The presence of gastrointestinal symptoms was assessed using a 16-item questionnaire. Participants used a 10-point Likert scale to rate whether they had experienced a gastrointestinal symptom (for e.g., reflux, heartburn, bloating, nausea, etc.) since bedtime the previous night. Responses ranged from 1, “no problem at all” to 10, “the worst it has ever been”.

### 3.8. Subjective Alertness and Self-Perceived Capacity

Subjective alertness was assessed using a visual analogue scale. Participants rated their current level of alertness by placing a vertical mark on a non-numeric, 100-mm line that was anchored with “not at all” at one end and “completely” at the other end. Self-perceived capacity was assessed using two separate visual analogue scales. Participants rated their ability to perform as fast as possible by placing a vertical mark on a non-numeric, 100-mm line that was anchored with “not fast at all” at one end and “very fast” at the other end. Participants rated their ability to perform as accurately as possible by placing a vertical mark on a non-numeric, 100-mm line that was anchored with “not accurately at all” at one end and “very accurately” at the other end.

### 3.9. Cognitive Performance

Sustained attention was assessed using the psychomotor vigilance task (PVT-192; Ambulatory Monitoring Inc., New York, NY, USA). The PVT is a hand-held device with an upper surface that contains a four-digit LED display and two push-button response keys. Participants attended to the LED display for the duration of the test (10 min) and pressed the appropriate response key with the thumb of their dominant hand as quickly as possible after the appearance of a visual stimulus (presented at a variable interval of 2–10 s). If the correct response key was pressed, the LED display exhibited the participant’s response time, in milliseconds, for 500 ms. If the wrong response key was pressed, an error message was displayed (ERR). If a response was made prior to the stimulus being presented, a false start message was displayed (FS). The dependent measures derived from the PVT included response time (ms); the number of lapses (i.e., response latency exceeding 500 ms) and the number of errors (i.e., false starts and incorrect button pushes). For all analyses, anticipated responses (i.e., those with response time less than 100 ms) were excluded. Cognitive throughput was assessed using a computer-based visual spatial-configuration search task [22,23]. The task is self-paced and consists of 52 trials in which participants are required to search for a target (i.e., the number 5) amongst distractors (i.e., the number 2). Visual searches were performed for set sizes of 10, 20, 30, or 40 distractor stimuli. Set size was equally distributed across the task. The dependent variables obtained from the task were the number of errors and the time taken to complete the task.

### 3.10. Postural Sway

Postural sway was assessed using an Accusway computerised force platform (AMTI, Watertown, MA, USA) in conjunction with Swaywin software (AMTI, Watertown, MA, USA). The force platform measures the three-dimensional forces (Fx, Fy, Fz) and the three-dimensional moments (Mx, My, Mz) involved in balance. These provide centre of pressure (COP) coordinates, which allow postural sway to be calculated. Participants performed two postural balance tasks each for 30 s; standing still with eyes open and standing still with eyes closed. The dependent variable obtained from the task was the area of the 95% confidence ellipse enclosing the COP.

### 3.11. Statistical Analysis

All data were analysed with a general linear mixed model using the R package lme4 (R Core Team). A random intercept for ‘subjects’ was included to account for intraindividual dependencies and interindividual heterogeneity. All models were estimated using restricted maximum likelihood. Data points with a value that was greater than 2 standard deviations from the mean were removed. Visual inspection of residual plots did not reveal any obvious deviations from homoscedasticity but showed indications of heavy tails against the normal distribution. This was accommodated by obtained bootstrapped confidence intervals. *p*-values were obtained using Type II Wald F tests with Kenward-Roger degrees of freedom as implemented in the R package. Results are reported as mean estimates and 95% confidence intervals

## 4. Results

### 4.1. Model Generation

Total tryptophan was dependent only on the amount of valerian and α-lactalbumin within the drink (Figure 1). The more of each of these nutrients, the greater the serum tryptophan levels.

A lower 3CVT score is an indicator of greater sleepiness. Contrary to tryptophan, 3CVT scores showed the most sensitivity to 5′AMP, theanine, and α-lactalbumin. For 3CVT, the most significant effect was from 5′AMP which showed a direct relationship between increasing levels of 5′AMP and decreasing 3CVT. As with tryptophan, increasing α-lactalbumin had a positive relationship with sleepiness (decreasing 3CVT); however, unlike tryptophan the effects of valerian were to decrease sleepiness (Figure 2).

When the Box–Behnken design was optimised for both tryptophan and 3CVT, a combination of 10 g high GI carbohydrate, 40 g α-lactalbumin, 655 mg theanine, 53 mcg 5′AMP, and 600 mg of valerian was predicted to be the best. This combination of ingredients was predicted to increase tryptophan to 2.25 µg/mL and decrease 3CVT score by 0.104 s. For the sake of validating the model in the subsequent sleep study, the worst combination and concentration of nutrients were predicted to be: 35 mL tart cherry, 45 g high GI carbohydrate, 8 g α-lactalbumin, 1000 mg theanine, 4.5 mcg 5′AMP, and 500 mg of valerian. These combinations are predicted to increase tryptophan 0.48 µg/mL and decrease 3CVT score by 0.001 s.

### 4.2. Sleep Study Validation

Objective and subjective sleep variables following drink consumption are reported in Table 1. Sleep onset latency was significantly lower in the optimal trial compared to both the placebo and least optimal trials (*p* = 0.02) (Figure 3D). None of the other variables (i.e., total sleep time, wake after sleep onset, sleep efficiency), were different between trials (Figure 3). Similarly, there was no change in the duration of time spent in sleep stages 1–3 or REM sleep (Table 1). Visual inspection of Figure 4 suggests that individuals with longer sleep onset latency respond in a more positive manner, relative to those with shorter sleep onset latency (Figure 4). However, this was not statistically significant as a consequence of the lack of power to address this question.

### 4.3. Gastrointestinal Symptoms, Self-Perceived Capacity, and Performance

There was no difference in next-day gastrointestinal symptoms, subjective alertness, self-perceived capacity, cognitive performance or postural sway between trials (Table 1).

## 5. Discussion

This study describes the use and validation of a predictive model to identify the optimal combination and dose of a suite of ingredients that have been independently suggested to improve sleep. Furthermore, this model was tested using polysomnography to determine the effects of the intervention on sleep quality and quantity. Overall, there was a 49% reduction in sleep onset latency following the consumption of the optimal drink compared to a placebo control drink. Interestingly, combining the same ingredients in the predicted least optimal manner resulted in a 33% increase in sleep onset latency when compared to the placebo control drink. These data suggest that the Box–Behnken model optimised for blood tryptophan and 3CVT was effective at predicting changes in sleep onset latency in human subjects. Importantly, the observed changes in sleep onset latency occurred despite all participants being considered normal or good sleepers. Furthermore, there were no adverse effects on sleep architecture and the optimal drink did not result in cognitive or balance impairments in the following morning. Of note, the improvements in sleep onset latency with the optimised intervention in the current study are similar to the improvements observed with pharmaceutical interventions [24].

Both tryptophan and high glycaemic index (GI) carbohydrate consumption prior to sleep are thought to influence neurotransmitters that are involved in the sleep–wake cycle [15]. Tryptophan shares blood brain barrier transporters with several large neutral amino acids, and an increase in the ratio of tryptophan to LNAAs may increase serotonin synthesis. Intake of a high GI carbohydrate may increase the Trp:LNAA in the brain via the insulin-stimulated [15] uptake of LNAA by muscle. Therefore, both α-lactalbumin (a milk protein containing high levels of tryptophan) and glucose supplementation prior to sleep have the potential to influence sleep.

Yajima and co-workers [25] investigated the effects of a high CHO, compared with a high fat, meal prior to sleep and reported a decrease in slow-wave sleep (Stage 3 or deep sleep) in the first half of the night in the high CHO trial. When low and high GI CHO-rich meals (consumed 4 h before bedtime) and a high GI meal (consumed 1 h prior) were compared, the high GI meal consumed 4 h before bedtime reduced sleep onset latency [26]. In a second study by the same authors [27], a very low CHO diet was compared to an energy-matched control diet, consumed 4 h prior to sleep. The very low CHO diet increased slow wave sleep and all stages of NREM sleep, whereas the control diet decreased REM sleep.

The potential sleep promoting ingredients investigated in the current study have been examined individually (i.e., not in combination with other ingredients) in a small number of studies. Research investigating the effects of tryptophan-rich protein on sleep indicates a reduction in time awake during the night, increased sleep efficiency, and increased subjective sleep quality [12]. When comparing α-lactalbumin to placebo (casein) prior to sleep, improved alertness the following morning has also been demonstrated [13]. The potential role for tryptophan in sleep is also supported by tryptophan depletion studies, which demonstrate an increase in sleep fragmentation [28].

Improved sleep quality and quantity has also been demonstrated with tart cherry juice consumption for 7 days [29]. Tart cherries have high concentrations of melatonin and other phenolic compounds which may have antioxidant and anti-inflammatory properties [16] that could potentially positively influence sleep. It should be noted that tart cherry juice may also contain high concentrations of carbohydrate, estimated to be approximately 30 g per 250 mL, which as mentioned above, may influence insulin concentrations.

Valerian (*Valeriana officinalis*) is one of the most commonly consumed ‘natural remedies’ for treating insomnia, and acts to inhibit sympathetic nervous system activity via the neurotransmitter GABA [30]. In a recent systematic review of various single plants to aid sleep, the authors concluded that due to different methods to measure sleep in the available literature (17 human trials) as well as varied doses used (225–1060 mg), studies demonstrated conflicting results. An earlier meta-analysis suggested that valerian may be effective for improving subjective sleep and is considered safe; however, there is insufficient objective data to determine its effectiveness.

L-Theanine is an amino acid found in tea which is reported to have anti-anxiolytic effects and induces changes in serotonergic and dopaminergic transmission [31]. Ingestion of 200 mg of L-Theanine has been shown to reduce wake after sleep onset, increase sleep efficiency, and have minimal daytime residual effects [31].

Nucleotides (purine adenosine 5′monophosphate (5′AMP), guanosine 5′monophosphate (5′GMP), and uridine 5′monophosphate (5′UMP) have been suggested to have a sleep-inducing role [14]. Specifically, the 5′AMP mechanism of action is thought to be via stimulation of the release of GABA [14]. The role of nucleotides in enhancing sleep has previously been investigated in infants. This is due to the higher concentrations of nucleotides in breast milk at night, demonstrating a circadian rhythm that was hypothesised to improve sleep. The minimal research in this area suggests that sleep was improved in infants who received a nucleotide-rich formula [32].

As a way to investigate all of the above-mentioned sleep promoting nutrients in one study, the current work tested all possible combinations of these six ingredients at three levels using an incomplete factorial design. This experiment would have required 729 different trials with groups of 10 or more participants per trial using traditional single factor methodology. Using a Box–Behnken incomplete factorial design, a model of the interactions between ingredients, and the necessity of each ingredient, was determined using only 48 trials. This model was then used to predict the optimal combination of ingredients based on both the maximal amount of tryptophan in the blood and the decrease in 3CVT measured following ingestion. A second combination of these same ingredients was established that was predicted to minimise blood tryptophan and increase 3CVT. This drink was predicted to negatively affect sleep and was used to validate the predictive capacity of the model. These combinations were then compared with a placebo control in the subsequent sleep studies.

The optimal combination of ingredients identified was a limited amount of glucose (10 g); the highest tested level of α-lactalbumin (40 g); no tart cherry juice (possibly due to the time of ingestion; 1 h prior to sleep); 600 mg of valerian (approximately mid-range based on previous research); a modest dose of theanine (655 mg); and the highest tested level of 5′AMP (53 mcg). The levels of α-lactalbumin and 5′AMP were highest in the optimal and lowest in the least optimal formulation suggesting that these ingredients have the strongest effect on the model. This is the first study of its kind to investigate this combination of ingredients, and there is limited research investigating the effects of more than one ingredient on sleep. Therefore, it is not possible to definitively describe the synergistic or potentially additive interaction of the ingredients. In the validation arm of the study, the optimal drink decreased sleep onset latency by 49% compared to the placebo control drink, whereas the least optimal combination of the same ingredients increased sleep onset latency by 33% compared to the placebo control drink. These results validate the predictive model and highlight the importance of understanding the precise combination and concentration of active ingredients within a supplement. Whether the effective concentration of the ingredients varies with body size is an important consideration that needs to be addressed in further research. Although not statistically significant, those participants who displayed the poorest sleep onset latency tended to show the greatest reduction in sleep onset latency with the optimal drink (Figure 4). This suggests that the optimal drink may be more efficacious for poor sleepers, or in individuals experiencing acute issues that impair sleep onset. However, both the optimal and least optimal drink had no effect on any other measure of sleep, which were all in the expected range for normal sleepers. Therefore, further research exploring the effects of this optimised sleep drink in poor sleepers, or in individuals who have difficulty initiating sleep onset, is warranted. Finally, an examination of the possible influence of this nutritional intervention on overnight muscle protein synthesis would be of interest. Previous work has demonstrated that protein ingestion prior to sleep can increase overnight muscle protein synthesis [33,34]. Furthermore, ingestion of protein prior to sleep was shown to be digested and absorbed throughout the night, muscle protein synthesis increased without disturbing sleep in both young and older healthy males [35,36]. If a nutritional intervention can both enhance sleep and improve overnight muscle protein synthesis, it may assist the recovery and performance of athletes.

In summary, an optimised combination of several nutritional ingredients, which had each demonstrated an influence on sleep independently, was determined. This intervention, when assessed utilising the gold standard of sleep monitoring in a controlled laboratory setting, significantly reduced sleep onset latency. Importantly, unlike pharmacological treatments, no detrimental effect on sleep architecture, subjective sleep measures, or next day performance were observed with the optimised nutritional supplement.

## Figures and Tables

**Figure 1 nutrients-12-02579-f001:**
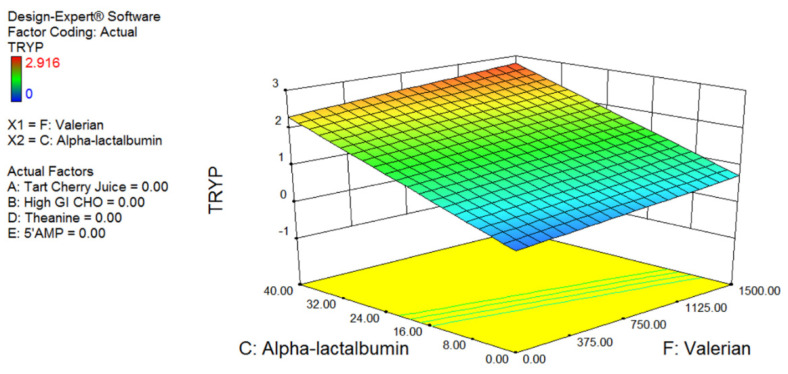
Quadratic response surface plot highlighting the effect of α-lactalbumin and valerian on blood tryptophan levels. The response surface represents all 55 measures. No other components affected this response surface.

**Figure 2 nutrients-12-02579-f002:**
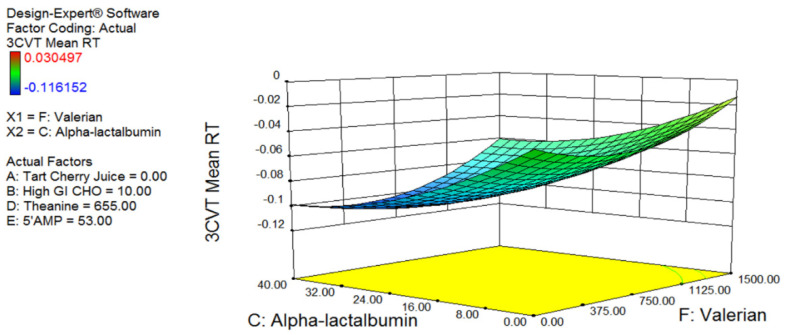
Quadratic response surface plot highlighting the effect of α-lactalbumin and valerian on three-choice vigilance task (3CVT) scores. A decrease in 3CVT score indicates greater sleepiness. The response surface represents all 55 measures. The 5′AMP linearly shifted the response down and is shown at its highest level. Theanine and high GI CHO had variable effects and are shown at their optimal.

**Figure 3 nutrients-12-02579-f003:**
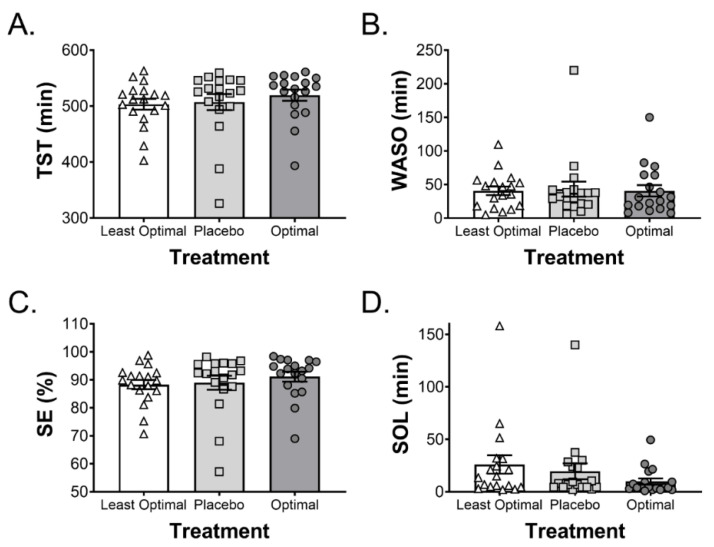
Individual data points for (**A**) total sleep time (TST), (**B**) wake after sleep onset (WASO), (**C**) sleep efficiency (SE), and (**D**) sleep onset latency (SOL). Bars and error bars are means and SD, respectively. Each point represents an individual.

**Figure 4 nutrients-12-02579-f004:**
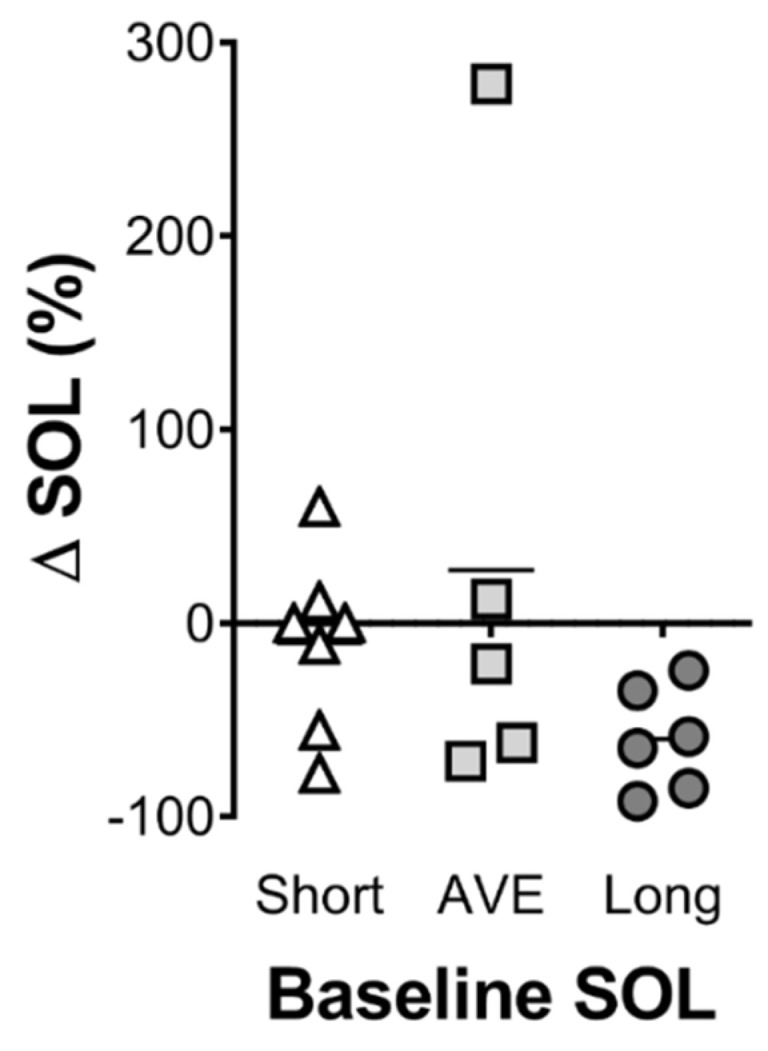
Baseline SOL and the magnitude of the effect of the drink. Individuals with the longest sleep onset latency (SOL) with the placebo treatment (23–140 min) tended to decrease sleep onset more than those with average (AVE; 7–15 min) or short (2–5 min) SOL when given the optimal drink. Lines are means and each dot is an individual.

**Table 1 nutrients-12-02579-t001:** Objective and subjective sleep variables and test battery results for least optimal, placebo, and optimal nutritional combinations (mean ± standard deviation (sd)).

	Least Optimal	Placebo	Optimal
	(Mean ± SD)	(Mean ± SD)	(Mean ± SD)
**Sleep Variables**			
Total Sleep Time (min)	503.3 ± 40.8	507.2 ± 60.5	519.5 ± 42.2
Wake after sleep onset (min)	40.6 ± 26.8	43.2 ± 47.3	40.6 ± 36.1
Sleep Efficiency (%)	88.3 ± 7.1	88.9 ± 10.6	91.1 ± 7.4
Sleep onset latency (min)	26.1 ± 37.4	19.6 ± 32.0	9.9 ± 12.3 *
REM Latency (min)	75.7 ± 25.6	90.3 ± 69.7	82.7 ± 33.3
Stage 3 Latency (min)	14.9 ± 6.8	14.8 ± 5.9	18.5 ± 13.3
Stage 1 (min)	29.1 ± 15.2	30.8 ± 11.6	32.7 ± 12.8
Stage 2 (min)	201.9 ± 43.8	204.8 ± 42.5	223.7 ± 44.9
Stage 3 (min)	143.2 ± 52.2	139.7 ± 49.5	135.4 ± 39.7
REM (min)	129.2 ± 208	131.9 ± 26.8	127.7 ± 29.3
Arousals—total (count)	98.5 ± 29.1	97.3 ± 25.8	106.1 ± 32.9
Arousals—REM (count)	24.2 ± 7.7	25.0 ± 11.1	23.0 ± 13.2
Arousals—NREM (count)	74.3 ± 32.6	72.3 ± 28.7	83.1 ± 36.1
Awakenings (count)	26.1 ± 7.9	24.7 ± 6.6	26.2 ± 8.3
Stage Shifts (count)	178.2 ± 46.5	173.4 ± 25.7	188.2 ± 38.0
KSS 2000 h (AU)	4.6 ± 0.9	4.3 ± 1.2	4.8 ± 0.9
KSS 2030 h (AU)	4.9 ± 1.0	4.8 ± 1.2	4.9 ± 1.0
KSS 2100 h (AU)	5.0 ± 1.1	5.3 ± 1.4	5.3 ± 1.1
KSS 2130 h (AU)	5.3 ± 1.2	5.9 ± 1.1	5.6 ± 1.0
KSS 2200 h (AU)	5.8 ± 1.1	6.1 ± 1.1	6.0 ± 1.1
Subjective Sleep Quality (AU)	4.6 ± 1.0	4.9 ± 1.1	4.7 ± 0.9
Subjective Sleep Quantity (h)	7.9 ± 1.4	8.0 ± 1.1	8.1 ± 1.0
Subjective SOL (min)	22.9 ± 17.8	18.8 ± 14.0	15.8 ± 9.7
**Test Battery**			
Mean Reaction Time (ms)	268.8 ± 49.6	260.5 ± 41.0	262.1 ± 39.2
Lapses (count)	1.6 ± 3.1	1.6 ± 2.8	1.4 ± 2.3
False Starts (count)	0.8 ± 1.0	0.8 ±1.3	0.5 ± 1.2
KSS 0900 h (AU)	4.6 ± 1.3	4.3 ± 1.3	4.3 ± 1.1
VAS Alertness 0900 h (AU)	57.4 ± 18.6	60.7 ± 20.3	63.1 ± 18.0
VAS Speed 0900 h (AU)	63.7 ± 14.9	65.9 ± 17.4	65.0 ± 16.1
VAS Accuracy 0900 h (AU)	66.6 ± 17.7	66.3 ± 19.4	67.4 ± 16.9
Postural Sway—Area 95 (cm^2^)	0.4 ± 0.2	0.4 ± 0.2	0.4 ± 0.4

* indicates significantly different from least optimal and placebo. SOL = sleep onset latency; REM = rapid eye movement; NREM = non rapid eye movement; KSS = Karolinska sleepiness scale; AU = arbitrary units; VAS = visual analogue scale.

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
