# Peer review of "Optimisation and Validation of a Nutritional Intervention to Enhance Sleep Quality and Quantity"

_nutrients, 2020, doi:10.3390/nu12092579_

Round 1

Reviewer 1 Report

Thank you for the opportunity to review this interesting paper with a novel approach to have drink formulation personalized to each participant. The below are my major comments.

Introduction

1. This section is relatively short and can include more backgrounds. For instance, in lines 60-61, each compound in the drink that investigated needs more background information, instead of leaving it out as it has some evidence that each compound influenced sleep. 

Methods

2. This is an intervention trial; however, many details are not included, such as dropout rates with reasons for the dropout and washout period between different combination/dose of the drink. Please refer to CONSORT or other clinical trial guidelines for other pieces of information to be included.

3. How the subjects in the validation study was selected? Were they representative of all subjects?

Results

4. Characteristics of participants before the trial were not well described. Age, height and weight were included, however, other information is not included. Other pieces of information are characteristics that may affect sleep such as occupation, total caloric and nutrient intakes, and lifestyle factors such as smoking, alcohol drinking and physical activity. In addition, their sleep characteristics before they started the trial need to be included.

5. In addition, due to effects of dietary compounds on sleep, what are participants’ total caloric and nutrient intakes during the intervention? It is mentioned that a standard diet was provided, but more information is needed.

6. Drink formulations were personalized. What are characteristics that are needed more of each compound (e.g., valerian)? For examples, those who are older tended to need more of valerian.

Discussion

7. It is novel that this study tested multiple ingredients shown to be effective for sleep in previous studies. However, there was no discussion of how multiple ingredients can exert synergetic effects from biological standpoints. Please add this to discussion.

8. Discussions of baseline characteristics other than SOL need to be included.

Author Response

Thank you for the opportunity to review this interesting paper with a novel approach to have drink formulation personalized to each participant. The below are my major comments.

We would like to thank the reviewer for their valuable comments. We have addressed each comment below.

Introduction

  1. This section is relatively short and can include more backgrounds. For instance, in lines 60-61, each compound in the drink that investigated needs more background information, instead of leaving it out as it has some evidence that each compound influenced sleep. 

Response: Some additional information has been added in the introduction, however there is a relatively large section providing evidence for each of the compounds in the discussion. The following has been added to the introduction:

A number of recent reviews have highlighted the potential for a number of these ingredients to influence sleep and jetlag (Doherty, 2019, Janse van Rensburg, 2020). However, each of these reviews make mention of the need for additional randomized controlled trials, using objective measures of sleep to enhance the quality of research in the area.

Methods

  1. This is an intervention trial; however, many details are not included, such as dropout rates with reasons for the dropout and washout period between different combination/dose of the drink. Please refer to CONSORT or other clinical trial guidelines for other pieces of information to be included.

Response: The project has been registered in the Australian New Zealand Clinical Trials Registry we are awaiting the notification of the approval number.

There were no participant dropouts and this statement has been added to the methods.

Line 119 contains the following information: The first night served as an adaptation night and was used to familiarize participants with the equipment for monitoring sleep. The next three nights were experimental nights on which participants received one of three interventions in a randomized, counterbalanced order: 1) optimal drink, 2) least optimal drink, or 3) placebo.

  1. How the subjects in the validation study was selected? Were they representative of all subjects?

Participants were enrolled in the study if they were within the designated age range, were healthy (no current medications) and no history of sleep disorders.

Results

  1. Characteristics of participants before the trial were not well described. Age, height and weight were included, however, other information is not included. Other pieces of information are characteristics that may affect sleep such as occupation, total caloric and nutrient intakes, and lifestyle factors such as smoking, alcohol drinking and physical activity. In addition, their sleep characteristics before they started the trial need to be included.

Response: The following information on participant characteristics has been added:

Participants were all non-smokers.  None of the participants had undertaken transmeridian travel or shift work in the month prior to participation in the study. Over the previous month, participants’ self-reported bedtime was 23:12 ± 00:54 h; self-reported get-up time was 08:12 ± 01:18 h; self-reported sleep onset latency was 15.0 ± 10.2 min and self-reported total sleep time was 8.1 ± 1.1 h.  Participants were excluded if their usual self-reported bedtime was later than 01:00h or their usual self-reported get-up time was later than 10:00 h.

Prior dietary intake was not accounted for as dietary intake was standardised and controlled for 24hrs prior to optimisation treatment ingestion and for the duration of the in-facility validation study period. The researchers believed that this combined with exercise control would be sufficient to standardise baseline nutrient availability in all subjects at the time of each individual treatment ingestion.

  1. In addition, due to effects of dietary compounds on sleep, what are participants’ total caloric and nutrient intakes during the intervention? It is mentioned that a standard diet was provided, but more information is needed.

Response: All meals were provided to the participants for the duration of the study. The following section from line 144 was provided:

All meals provided to participants were calorie-controlled and the participants consumed the same approximate number of calories at each meal. The participants were served breakfast, lunch, and dinner at 09:30h, 12:30h and 19:00h, respectively. An afternoon snack was also provided at 16:00h. On average, participants consumed 10,061 ± 1112 kJ per day containing 51.8 ± 6.2% carbohydrate, 21.7 ± 6.5% protein, and 23.8 ± 6.7% fat.   Participants did not have access to any other food or beverages (other than water) outside of the designated meal/snack times and were not permitted to consume caffeine or alcohol at any time during the protocol.

  1. Drink formulations were personalized. What are characteristics that are needed more of each compound (e.g., valerian)? For examples, those who are older tended to need more of valerian.

Response: The drink formulations were not individualised. As mentioned from line 122: The next three nights were experimental nights on which participants received one of three interventions in a randomized, counterbalanced order: 1) optimal drink, 2) least optimal drink, or 3) placebo.

Age was not included as a covariate due to the homogenous nature of the participants.

 Discussion

  1. It is novel that this study tested multiple ingredients shown to be effective for sleep in previous studies. However, there was no discussion of how multiple ingredients can exert synergetic effects from biological standpoints. Please add this to discussion.

Response: The following has been added to the discussion: This is the first study of its kind to investigate this combination of ingredients and there is limited research investigating the effects of more than one ingredient on sleep. Therefore, it is not possible to definitively describe the synergistic or potentially additive interaction of the ingredients.

  1. Discussions of baseline characteristics other than SOL need to be included.

Response: The following has been added to the discussion: However, both the optimal and least optimal drink had no effect on any other measure of sleep, which were all in the expected range for normal sleepers.

Reviewer 2 Report

The manuscript has been written in a good order and the research outcomes are clear.

I have some questions need to be respond by the author.

  1. Ethic concern: Dose this study has been approved by IRB (Institutional Review Board)? The author should declare this issue in the part of study design and Ethic Approval. The protocol should be reviewed by IRB and all participates should have informed consent prior to joint the current study.
  2. In the first part of study regarding optimization of nutritional intervention, 55 healthy males have been enrolled into the study. How did you calculate the sample sizes of study? Why the study just enrolled male patients?
  3. In Lin 76-79. “According to the Box-Behnken design, each subject was provided with one intervention, making for 55 independent trials (48 intervention trials, plus 7 centre point trials)…..”. Could you explain the difference between 48 intervention trials & 7 center point trials? How did you select subject into intervention trials or center point trials? Since the Box-Behnken design may not be familiar with the most readers, suggest the author try to make it simple and clear.
  4. In the second part of study regarding validation the model by enrolling 18 healthy males, normal sleepers into the study. How did you calculate the sample sizes of study? The definition of normal sleepers is subjective or objective according to prior PSG test?
  5. The only positive findings regarding the effect of nutritional ingredients on PSG is sleep onset latency in the current study. The optimal drink group revealed a 49% reduction in sleep onset latency, however, the least optimal drink resulted in a 33% increase in sleep onset latency. Since this result is contributed from the different component of ingredients, suggest make a short summary in the part of discussion to compare the difference of these two groups.

Author Response

The manuscript has been written in a good order and the research outcomes are clear.

We would like to thank the reviewer for their valuable comments. We have address ed each comment below.

I have some questions need to be respond by the author.

  1. Ethic concern: Dose this study has been approved by IRB (Institutional Review Board)? The author should declare this issue in the part of study design and Ethic Approval. The protocol should be reviewed by IRB and all participates should have informed consent prior to joint the current study.

Response: The following was added to line 83:

The study was approved by the Australian Institute of Sport Ethics Committee (Approval number: 20131003) and all participants completed written informed consent. The study was registered on the Australian New Zealand Clinical Trials Registry (tbc).

2.In the first part of study regarding optimization of nutritional intervention, 55 healthy males have been enrolled into the study. How did you calculate the sample sizes of study? Why the study just enrolled male patients?

Response: In a bid to compare dosages and timing to other research investigating these nutritional ingredients, male participants were chosen. A series of follow up studies are planned that will include female participants.

3.In Lin 76-79. “According to the Box-Behnken design, each subject was provided with one intervention, making for 55 independent trials (48 intervention trials, plus 7 centre point trials)…..”. Could you explain the difference between 48 intervention trials & 7 center point trials? How did you select subject into intervention trials or center point trials? Since the Box-Behnken design may not be familiar with the most readers, suggest the author try to make it simple and clear.

Response: The following has been added to the text in line 81: The 7 centre point trials consisted of participants being provided the mid-range dose for each ingredient.

The following has been added to line 84: As each participant completed one trial only, the subjects were allocated a supplement in the order they were recruited.

4.In the second part of study regarding validation the model by enrolling 18 healthy males, normal sleepers into the study. How did you calculate the sample sizes of study? The definition of normal sleepers is subjective or objective according to prior PSG test?

Response: Participant numbers was based on previous research conducted examining nutritional interventions in sleep. Further, increased numbers (n=18) were recruited as the sleep laboratory is a 6 bed facility, there were 3 interventions and this allowed for appropriate allocation to groups.

5.The only positive findings regarding the effect of nutritional ingredients on PSG is sleep onset latency in the current study. The optimal drink group revealed a 49% reduction in sleep onset latency, however, the least optimal drink resulted in a 33% increase in sleep onset latency. Since this result is contributed from the different component of ingredients, suggest make a short summary in the part of discussion to compare the difference of these two groups.

Response:

Based on requests from both reviewers, the following has been added to the discussion:

Line 367: This is the first study of its kind to investigate this combination of ingredients and there is limited research investigating the effects of more than one ingredient on sleep. Therefore, it is not possible to definitively describe the synergistic or potentially additive interaction of the ingredients.

Line 380: However, both the optimal and least optimal drink had no effect on any other measure of sleep, which were all in the expected range for normal sleepers.

Round 2

Reviewer 1 Report

Thank you for the opportunity to review the revised version of this interesting manuscript. The authors have addressed some of my previous comments in the revised manuscript, but not others as below (comment numbers in the previous review are included for your convenience). Please address these comments fully in the revised manuscript.

Methods

  1. Washout period

Thank you for including this information. The revised manuscript is still missing is the rationale of conducting three different drink formulations on three consecutive days. What evidence have the authors used to decide that this is sufficient to address washout period, based on compounds they are tested?

Results

  1. Characteristics of participants before the trial

How about the alcohol/ethanol consumption?

  1. Nutrient contents of the standard meals provided during the trial.

The authors have provided information on macronutrients, but not micronutrients (vitamins and minerals). Please add this information in the next revision. Given that micronutrients regulate energy metabolism and that the authors mention the importance of understanding active ingredients (that do not contribute to energy) in Discussion section, readers and researchers who plan future studies based on your study finding will benefit from this information.

Discussion

  1. It is understood that there is no definitive evidence on the specific combinations this trial has tested. However, it is very important and beneficial for readers and researchers to have the information on what we know, even it is not definitive. It is relevant to include evidence from in vitro or animal studies, as needed. Going back to the rationale of conducting this trial, the novelty was testing multiple ingredients. However, why it is important to do this was not explained (other than it has not been tested). For molecular mechanisms/pathways, the authors have mentioned multiple neurotransmitters and related intermediate compounds (mechanism 1) and energy/macronutrient metabolism (mechanism 2); however, how multiple compounds can play a role in mechanism 1, mechanism 2, etc are not mentioned. Please include one paragraph describes for each mechanism, which compounds in the drink formulations may have contributed. This will help readers clarify on which mechanisms were more important than others in your trial.

  1. My comment was on any effect of non-sleep characteristics such as age and body size. The authors mention in lines 382-383: “Whether the effective concentration of the ingredients varies with body size is an important consideration that needs to be addressed in further research.” Given that the authors have collected information on body size, it is possible to address this by conducting analysis separately by small vs large small size. Please include this in Results and Discussion section in the next revision.

Author Response

We thanks the reviewer for the constructive comments and feedback

Reviewer 2 Report

All the questions has been responded by the author adequately.

Author Response

Thank you for your comments and feedback